

# The relationship between adolescent obesity and pelvis dimensions in adulthood: a retrospective longitudinal study

Jan M. Novak[1], Jaroslav Bruzek[1], Hana Zamrazilova[2], Marketa Vankova[2], Martin Hill[2] and Petr Sedlak[1]

[1] Department of Anthropology and Human Genetics, Faculty of Science, Charles University, Prague, Czech Republic
[2] Institute of Endocrinology, Prague, Czech Republic

Corresponding author
Jan M. Novak,
janmnovak@gmail.com

## ABSTRACT

**Background:** The effect of fat tissue on a developing individual is fundamentally different from the effect on an adult. Several changes caused by obesity during sexual maturation have an irreversible and severe negative effect (lower fertility, reduced final height, type 2 diabetes mellitus) even for those who have subsequently lost weight. Our study was focused on monitoring the skeletal structure substantially influenced by sex hormones—the pelvis. The adult pelvis is strongly sexually dimorphic, which is not the case for the juvenile pelvis; skeletal differences between sexes are not so prominent and start to manifest with the onset of puberty. Evidence from animal models and case studies of treatment of gender dysphoria suggests that estrogens have a stimulatory effect on the growth plates present on the pelvis, leading to morphological change. Male obesity, especially in puberty, is connected with hypogonadism, manifesting in low levels of testosterone, and high levels of estrogens. The goal of our study was to evaluate the influence of obesity during adolescence on the morphology of the adult pelvis in the context of androgen and estrogen status.

**Sample and Methods:** Our sample consists of 238 individuals (144 females, 94 males) observed after an 8 year follow-up (mean age during enrollment 15.2 years, follow-up 23.3 years). Anthropometry and body composition using bioimpedance analysis (BIA) were obtained. During the follow-up, saliva samples from male participants were also collected to estimate testosterone and estradiol levels using the salivary ELISA kit (Salimetrics LLC, State College, PA, USA).

**Results:** The body fat (percentage of body fat estimated using BIA) was strongly positively associated with relative pelvic breadths in adulthood (males $r = 0.64$; females $r = 0.56$, both with $p < 0.001$). Adulthood pelvic breadth was a highly sensitive (0.81) and specific (0.74) retrospective marker of obesity during adolescence. The complex regression model (with reduction of dimensionality) including testosterone, estradiol to testosterone ratio and body fat (adolescent and adulthood) was able to describe 54.8% variability of pelvic breadth among males.

**Discussion:** We observed that adults with a history of obesity from adolescence tend to have a wider dimension of the bony pelvis in adulthood. Based on the parameters of the adult pelvis, the history of obesity can be determined with satisfactory

sensitivity and specificity (<70%). One of the explanations for this observation can be the influence of relatively elevated estrogens levels connected with excessive adiposity leading to a wider pelvis. However, the biomechanical stress connected with elevated body mass also has to be considered, as does the influence of physical activity and gait pattern on the skeletal build.

## INTRODUCTION

Bone tissue is highly plastic and capable of a wide degree of adaptability in response to biomechanical factors, and at the same time, its development and structure are dependent on nutrition and hormonal regulation (*Frost, 2000*; *Devlin, 2011*). Modern lifestyle represented by hypokinesia and excessive caloric intake results in an increasing prevalence of obesity among adults and children. The impacts of obesity on skeletal parameters have been studied mostly from the perspective of changes in bone density, mass and strength (*Farr & Dimitri, 2017*; *Dimitri, 2019*). The evidence of changes in skeletal morphology in the context of obesity is limited, especially in the case of juvenile individuals.

The World Health Organization (WHO) defines obesity as abnormal or excessive fat accumulation posing a risk to health (*WHO, 2018*). The model of fat tissue as inactive energy storage has been challenged in the past years. Current research indicates that excessive secretion of pro-inflammatory adipokines in fat tissue leads to chronic inflammation and other severe comorbidities, the most important ones being hypertension and other cardiovascular disorders, insulin resistance, type 2 diabetes mellitus, dyslipidemia, atherosclerosis and depression (*Lean, 2000*; *Redinger, 2007*; *Taylor & MacQueen, 2010*).

The negative influence of obesity in the period of rapid development is different from that in adulthood; this influence is often severe and irreversible. Excellent examples include reduction of final height of obese girls related to an earlier onset of puberty (*Li et al., 2017*), lower fertility in adulthood (*He et al., 2018*), higher nulliparity rates among females with history of obesity from adolescence (*Polotsky et al., 2010*), as well as a higher risk of type 2 diabetes mellitus and cardiovascular disorders (*Ayer et al., 2015*). Interestingly, there is well reported evidence that the risk of many disorders in adulthood (colorectal cancer, atherosclerosis, gout) is more dependent on the history of obesity from adolescence than the current obesity status (*Must et al., 1992*).

Not all pathogenic effects of fat tissue may be explained just by excessive secretion of adipokines; lower final height and lower fertility are also connected with another effect of fat tissue: estrogenization. Estrogenization, that is, the exposure of an organism to estrogens, is to a certain degree normal for both sexes. It is necessary to highlight here that estrogens are not female-only hormones (*Berthrong, Goodwin & Scott, 1949*). Estradiol in males is involved in several biological and behavioral processes such as

spermatogenesis, regulation of libido and erection (*Schulster, Bernie & Ramasamy, 2016*). For premenopausal females, the main sources of estrogens are ovaries, but in the case of postmenopausal females and males in general, estrogens are synthesized extragonadal on the local level (*Simpson, 2003*; *Labrie et al., 2017*). The same enzyme that catalyzes the transformation of delta-4 C19 steroids (predominantly androstenedione and testosterone) into estrogens in the ovary tissue (aromatase cytochrome P450) is relatively highly concentrated in fat tissue (*Agarwal et al., 1997*). In addition, inflammation caused by adipokines leads to even higher aromatase concentration in fat tissue (*Polari et al., 2015*; *Iyengar et al., 2017*). The amount of fat tissue was positively correlated with the level of estrogens in observational studies on males (*Gettler et al., 2014*) as well as females (*Ziomkiewicz et al., 2008*). Because the main source of estrogens is in this case conversion (depletion) of circulating testosterone, its levels are reduced and expressed with a higher E2/T (and vise versa) ratio (*Kelly & Jones, 2015*). Fat and ovaries are not the only tissues with high expression of aromatase; brain, periosteum, prostate and placenta are also relatively rich in aromatase (*Stocco, 2012*), but the fat tissue is most variable tissue in amount and probably best proxy to general aromatase activity among males and postmenopausal females.

In the context of obesity, estrogens have one remarkable effect: they lead to increased deposition of fat tissue, that is, estrogens are obesogenic and obesity itself is pro-estrogenic. This positive feedback loop has already been described in the example of the development of obesity-related male hypogonadism (*Cohen, 2007*; *Kelly & Jones, 2015*). The obesogenic activity of estrogens can explain the overall higher fat content in females; the rapid fat gain after the onset of puberty observed among girls is probably caused by elevated levels of estrogens (*Grantham & Henneberg, 2014*).

In terms of human growth, estrogens have been studied for their stimulatory effect on epiphyseal growth plates of long bones and the related influence on body height and also stimulatory effect on periosteal growth and overall skeletal robusticity (*Weise et al., 2001*; *Devlin, 2011*). Obesity-related estrogenization leads to premature acceleration of growth in both sexes and acceleration of sexual development among females resulting in lower adult height (*Brener et al., 2017*). The evidence regarding the acceleration effect of obesity on sexual development in males is ambivalent (*Wagner et al., 2012*; *De Leonibus, Marcovecchio & Chiarelli, 2012*). Long bones are not the only skeletal structure influenced by estrogens. The morphology of the adult bony pelvis is highly sexually dimorphic in comparison to the juvenile pelvis. Many parameters on the juvenile pelvis are not fully manifested and morphological differences are not so prominent as in adulthood (*Coleman, 1969*; *Huseynov et al., 2016*). One of these parameters is the pelvic breadth and pelvic breadth in relation to body height (relative pelvic breadth).

The first signs of sexual difference in pelvic breadth start to manifest with the onset of puberty in girls (*Goyal, 2016*). One of the discussed factors is the stimulatory effect of female sex hormones on the growth plates present in the pelvis (*Huseynov et al., 2016*). The pelvis-widening effect of the administration of estrogen analogs has been observed in animal models (*Hughes & Tanner, 1974*). In contrast, the pelvis morphology of estrogen-deficient women is described as male-android (*Witchel, 2012*). Remarkably, the

development of bony pelvis parameters does not strictly end with maturity, although the period of childhood and adolescence are to most dynamic (*Huseynov et al., 2016*). The evidence suggests that the most rapid and striking development of pelvic breadth ends between 20 and 21 years of life, followed by a gradual slow increase up to 60 years of age observed in a Japanese population of both sexes (*Ikoma et al., 1988*). The study on Czech males found the even earlier end of excessive adolescent growth between 17 and 18 years for the pelvic breadth and pelvic breadth in relation to body height (*Parizkova, 1970*).

The present study aims to investigate the potential influence of fat tissue in adolescence on the skeletal build in adulthood. We monitored skeletal changes on the pelvis as the highly sexually dimorphic parameter under the direct control of sexual hormones. We assume that individuals who had higher body fat during adolescence were also more estrogenized and tend to have a relatively broader pelvis. If the previous assumption is valid, we want to test the possible validity and application of the relative pelvic width as a classifier of history of obesity and a marker of the current androgenic status. Based on these assumptions and aims, we formulated the following hypotheses:

1. Adiposity (the body fat %) from adolescence is positively related to relative pelvic breadth in adulthood.
2. Inversely, the history of obesity from adolescence can be precisely distinguished using relative pelvic breadth in adulthood.
3. The relative pelvic breadth is negatively related to the androgenic status (based on salivary testosterone and estradiol to testosterone ratio).

## MATERIALS AND METHODS

The study sample involved longitudinally monitored Czech individuals (European ancestry) examined twice: in adolescence (mean age 15.2, SD 1.58; enrollment) and after a period of 8 years in early adulthood (mean age 23.3, SD 2.01; follow-up). The first round of examination (enrollment) was conducted during 2010, the second examination (follow-up) during 2018. We analyzed only probands who completed both (adolescent and adulthood) monitoring examinations; with the total $n$ of probands being 238 (144 females, 94 males). The complex examination included standardized anthropometry, estimation of the percentage of body fat (BF%) using bioelectrical impedance analysis (BIA) (Tanita BC-418 MA, Tanita SC-240 MA) and levels of testosterone (T) and estradiol (E2) using researcher-grade salivary ELISA kits (Salimetrics LLC, State College, PA, USA) among male participants. The collection of saliva samples was performed in standardized conditions; participants were instructed to not eat or drink anything than water after waking up. Considering the diurnal rhythms in steroid hormones concentrations (*Brambilla et al., 2009*) and the requirement for fasting, all male participants were examined between 8:00 AM and 10:00 AM. Saliva samples were collected using a passive drool method with the collection aid SB-WS (Salimetrics LLC, State College, PA, USA). Samples were stored at −80 °C until analyzed. A total of 11 participants were not able to provide saliva samples, six were excluded during testosterone analysis and

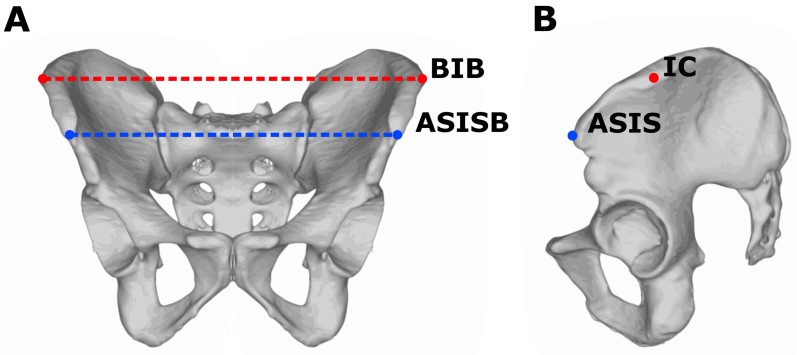

**Figure 1 Measurements of pelvic breadth.** Frontal (A) and lateral (B) view on the pelvis. The red line represents bi-iliac breadth (BIB) defined as the distance between most lateral points (IC) on the outer edge of the iliac crest. The blue line represents bi-spinal breadth (ASISB) defined as the distance between the left and right anterior superior iliac spine (ASIS).

an additional five during estradiol analysis due to nonstandard pH values in accordance with the protocol provided by the kit manufacturer. Also, the concentration of estradiol of 6 participants was below the detection range of the kit.

The pelvic breadth was measured using a pelvimeter (Trystom P-216, Klášterní Hradisko, Olomouc, Czechia); the pelvic breadth was defined as the distance between most lateral points on the outer edge of iliac crest, that is, bi-iliac breadth (BIB). The measurement of BIB can be theoretically influenced by the accumulation of soft tissues. The bi-spinal breadth defined as the distance between the left and right anterior superior iliac spine (ASISB) was obtained as the control parameter. The advantage of bi-spinal breadth over BIB is that there is no interaction with soft tissues during the measurement, as the ASISB is anteriorly prominent bony landmark (Fig. 1). Both pelvic dimensions were analyzed in relation to body height (selected pelvic breadth in cm/body height in cm × 100) and indicated as RBIB (relative BIB) and RASISB (relative bi-spinal breadth). The body height was measured using a portable stadiometer (GPM model 100) in a standardized upright position with the head positioned in the horizontal Frankfurt plane and defined as the distance between vertex point and floor (*Martin & Saller, 1957*; *Eston & Reilly, 2009*). Obesity in adolescence was classified as the BMI Standard Deviation Score (BMI-SDS) higher than 2 SD, using age and sex-specific national Czech reference sample (*Kobzová et al., 2004*). BMI-SDS methods (also known as *Z*-score method) can reflect dynamic changes in BMI during childhood and adolescence (*Cole, 1990*; *Cole et al., 2000*; *Must & Anderson, 2006*). Obesity in adulthood was defined as BMI above 30 kg/m$^2$. All participants were acquainted with the study aims and methods and gave written informed consent to participate. The study was approved by the institutional review board of the Faculty of Science, Charles University in Prague (IRB Approval No. 201719).

## Statistical analysis

Statistical analysis was conducted in the R Programming Language version 3.5.1 (*R Development Core Team, 2018*) and SIMCA version 12 (MKS Umetrics AB, Umeå, Sweden).

In R were used following packages: tidyverse group of packages (*Wickham, 2017*) and pROC (*Robin et al., 2011*). The relationships between parameters and groups were tested using standard methods such as linear regression, Mann–Whitney $U$ test, Spearman's rank-order correlation (analyzed data had non-normal data distribution), multivariate regression with a reduction of dimensionality using the method of orthogonal predictions to latent structure (OPLS) (*Trygg & Wold, 2002*) and ordinary multiple regression. The OPLS model is capable to cope with severe multicollinearity in the set of predictors in contrast to ordinary multivariate regression (*Vajargah et al., 2012*). For the validation of classification properties, the Receiver Operating Characteristic Curves (ROC) analysis was used (*Fawcett, 2006*), the optimal cutoff value was calculated using Youden index (*Youden, 1950*). Unless otherwise stated, curves in plots are construed using the locally estimated scatterplot smoothing method (*Savitzky & Golay, 1964*; *Cleveland, 1979*; *Wickham, 2009*). The univariate data distribution and data homogeneity were tested using diagnostic plots and criteria and data were transformed toward normality if that was required for regression using Box–Cox Power transformation (*Box & Cox, 1964*). The multivariate data distribution and data homogeneity were tested using the tools in the SIMCA (*Trygg & Wold, 2002*). The variables with skewed data distribution were transformed to attain symmetric distribution and homoscedasticity (constant variance) (*Meloun et al., 2000*). All data including error values and outliers are publicly accessible (Data Supplement).

# RESULTS

## Descriptive statistics

The prevalence of obesity determined using BMI (BMI > 30; respectively BMI-SDS > 2 in adolescence) was 22% during enrollment in adolescence (23% in females, 21% in males) and 19% during follow-up in adulthood (17% in females, 22% in males). An observed relative high prevalence of obesity was in accordance with obesity-targeted study design. In respect of longitudinal changes in obesity status, most participants that had non-obese BMI-SDS values in adolescence remained with non-obese values as adults ($n$ = 171, 72% of all participants), and lower fraction converted to obesity ($n$ = 21, 9% of all participants); the majority of individuals positively screened for obesity in adolescence also remained obese in adulthood ($n$ = 32, 13% of all participants), but almost one third ($n$ = 14, 6% of all participants) were able to lose their weight and move out of the obesity range.

The main characteristics and sexual differences of the study sample are shown in (Table 1).

## Relationship of pelvic breadth and body fat

Adult values of RBIB were strongly correlated with BF% in adolescence among both males ($r$ = 0.64, $p$ < 0.001) and females ($r$ = 0.56, $p$ < 0.001) (Fig. 2). The relationship was stronger for adult BF% among males ($r$ = 0.80, $p$ < 0.001) and females ($r$ = 0.73, $p$ < 0.001) (Fig. 3).

**Table 1 Sample characteristics.** Shows sample characteristics and significance of their sexual difference (Mann–Whitney $U$ test).

| | Variable | Females | | Males | | Sexual difference | |
|---|---|---|---|---|---|---|---|
| | | Median (quartiles) | Mean ± SD | Median (quartiles) | Mean ± SD | Mann–Whitney $U$ | $p$-value |
| Adolescence | Age (years) | 14.8 (13.8, 16.4) | 15.1 (1.66) | 15.1 (14, 16.6) | 15.3 (1.46) | 5,988 | 0.343 |
| | BMI (kg/m$^2$) | 21.9 (19.6, 26.2) | 23.1 (4.71) | 21.9 (19.4, 25.7) | 23.2 (4.9) | 6,438 | 0.960 |
| | BMI-SDS | 0.8 (−0.22, 1.93) | 0.79 (1.38) | 0.69 (−0.11, 1.74) | 0.78 (1.34) | 6,311 | 0.904 |
| | Body fat—BIA (%) | 25.6 (22.3, 30.9) | 27.2 (7.22) | 17 (14.1, 23.4) | 19.7 (7.56) | 2,666 | <0.001** |
| Adulthood | Age (years) | 23 (21.6, 24.8) | 23.2 (2.07) | 23.9 (21.5, 25.2) | 23.5 (1.96) | 7,049 | 0.361 |
| | RBIB | 16.9 (16.1, 18.1) | 17.4 (1.86) | 16.4 (15.7, 17.3) | 16.6 (1.3) | 4,968 | <0.001** |
| | RASISB | 15.2 (14.2, 16.3) | 15.5 (1.73) | 14.5 (13.9, 15.4) | 14.7 (1.0) | 4,739 | <0.001** |
| | BMI (kg/m$^2$) | 23 (20.6, 27.6) | 25.3 (6.51) | 25.5 (23, 28.8) | 26.4 (5.37) | 5,471 | 0.012* |
| | Body fat—BIA (%) | 27.7 (22.6, 35.5) | 29.4 (8.88) | 17.9 (14.1, 22.9) | 18.9 (7.07) | 2,270 | <0.001** |
| | Salivary T (pg/mL) | – | – | 188 (155, 226) | 203 (78.1) | – | – |
| | Salivary E2 (pg/mL) | – | – | 1.4 (0.98, 1.73) | 1.38 (0.54) | – | – |
| | E2/T × 1,000 | – | – | 6.38 (4.92, 9.38) | 7.43 (4.17) | – | – |

Notes:
* $p < 0.05$.
** $p < 0.01$.

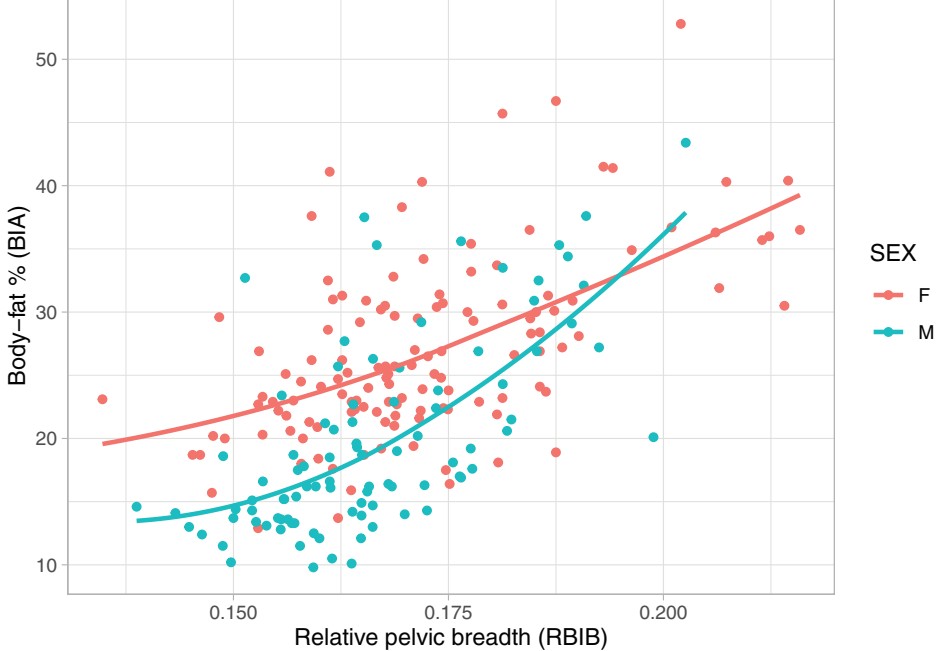

**Figure 2 Relationship between body fat percentages (BIA) in adolescence and relative pelvic breadth in adulthood (RBIB).** The $X$-axis represents adulthood relative pelvic breadth (bi-iliac breadth to body height), the $Y$-axis represents body fat percentages in adolescence estimated using BIA. Group M (males) $r^2 = 0.39$, Group F (females) $r^2 = 0.33$, both on $p < 0.001$.

The relationship between BF% and RASISB was comparable to RBIB. Adulthood values of RASISB were correlated to BF% in adolescence among males ($r = 0.58$, $p$ value <0.001) and females ($r = 0.53$, $p$ value <0.001). Similiary to RBIB, the relationship

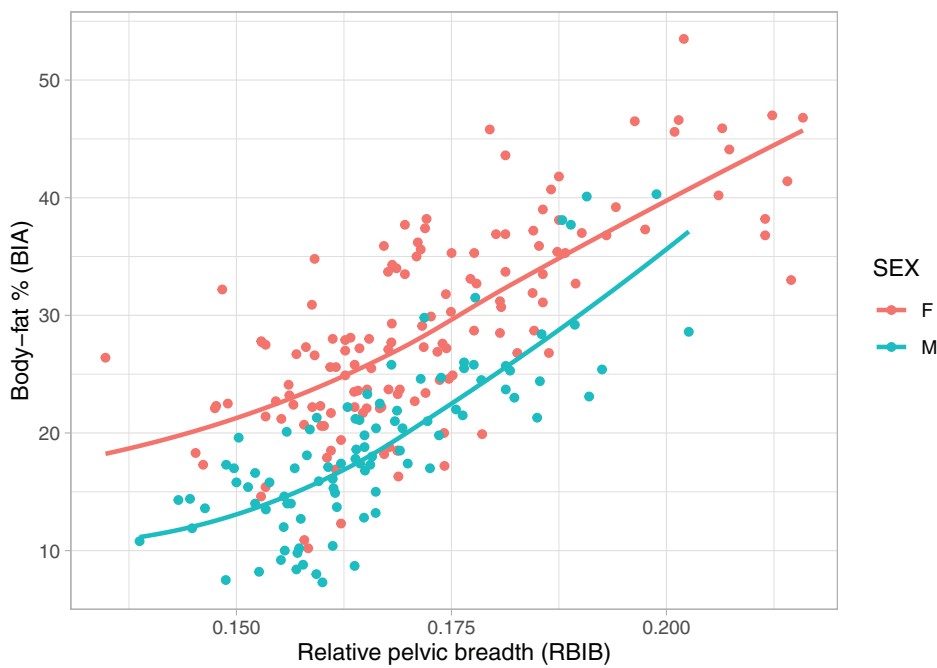

**Figure 3 Relationship between body fat percentages (BIA) and relative pelvic breadth in adulthood (RBIB).** The $X$-axis represents adulthood relative pelvic breadth (bi-iliac breadth to body height), the $Y$-axis represents body fat percentages in adulthood estimated using BIA. Group M (males) $r^2 = 0.54$, both on $p < 0.001$, Group F (females) $r^2 = 0.50$.

was stronger for adulthood BF% among both females ($r = 0.68$, $p < 0.001$) and males ($r = 0.65$, $p < 0.001$).

## Pelvic breadth as a discrimination factor of obesity

The values of BMI (respectively BMI-SDS in adolescence) were significantly correlated to the BF% during adolescence ($r = 0.76$, $p < 0.001$) and adulthood ($r = 0.62$, $p < 0.001$).

Adult individuals with a history of obesity from adolescence manifested significantly higher values of RBIB, both males (males with history of obesity $n = 20$, mean = 17.99, SD = 1.25, males without history of obesity $n = 74$, mean = 16.19, SD = 1.03; Mann–Whitney $U = 178$, $p < 0.001$) and females (females with history of obesity $n = 33$, mean = 18.98, SD = 2.20, females without history of obesity $n = 111$, mean = 16.87, SD = 1.43, Mann–Whitney $U = 725$, $p < 0.001$).

RASISB manifested same trend in difference between individuals with and without history of obesity as RBIB. Both males (males with history of obesity $n = 20$, mean = 15.75, SD = 1.11, males without history of obesity $n = 74$, mean = 14.45, SD = 0.90; Mann–Whitney $U = 276$) and females (females with history of obesity $n = 33$, mean = 17.07, SD = 2.11, females without history of obesity $n = 111$, mean = 15.11, SD = 1.30, Mann–Whitney $U = 744$, $p < 0.001$) with history of obesity from adolescence had significantly higher values of RASISB.

RBIB in adulthood was a sensitive and specific retrospective marker of history of obesity in adolescence. The discrimination rates were lower for females (sensitivity = 0.85,

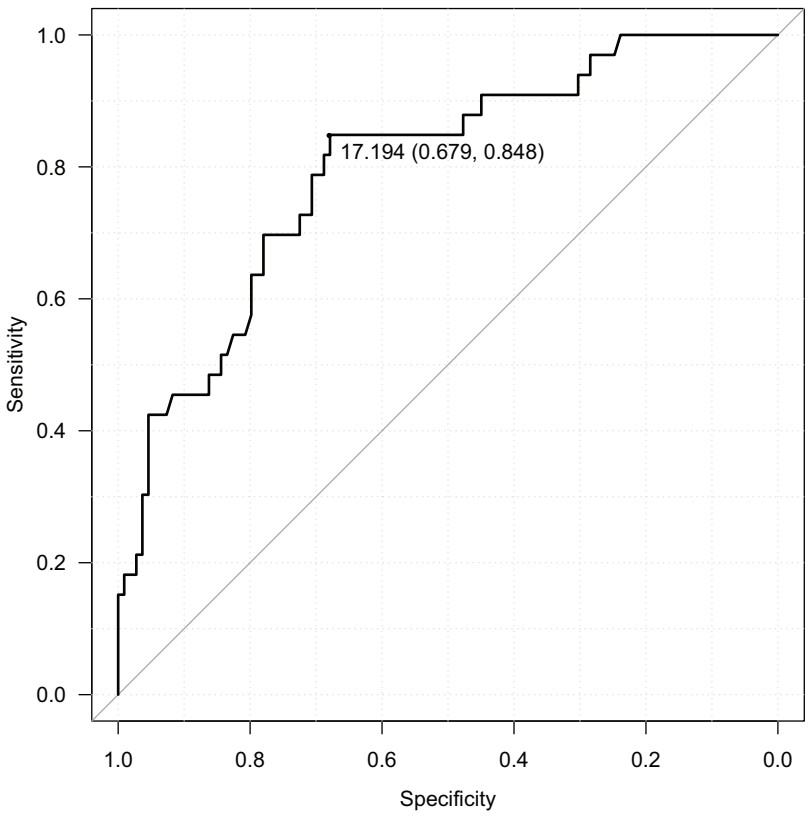

**Figure 4 ROC curve of adulthood RBIB discrimination proprieties of adolescent obesity (females).**
The black dot represents an optimal RBIB cutoff value for distinguishing the history of obesity (17.2) with
high sensitivity (0.85) and specificity (0.68).

specificity = 0.68, Area Under Curve = 0.80; using cutoff value 17.19) (Fig. 4) than
males (sensitivity = 0.80, specificity = 0.81, Area Under Curve = 0.87; using cutoff value
16.93) (Fig. 5). Even without distinguishing sex, RBIB provided a relatively high
discrimination rate of obesity during adolescence (sensitivity = 0.81, specificity = 0.74,
Area Under Curve = 0.82; using cutoff value 17.19) (Fig. 6).

RASISB (control parameter) in adulthood was also a sensitive and specific retrospective
marker of history of obesity in adolescence. Similarly to RBIB, the discrimination rates
were lower for females (sensitivity = 0.73, specificity = 0.78, Area Under Curve = 0.79;
using cutoff value 16.02) than males (sensitivity = 0.65, specificity = 0.93, Area Under
Curve = 0.83; using cutoff value 15.59). Without distinguishing sex, RASISB also provided
a relatively high discrimination rate of obesity during adolescence (sensitivity = 0.74,
specificity = 0.78, Area Under Curve = 0.80; using cutoff value 15.59).

## Pelvic breadth as a marker of adipose estrogenization among males

RBIB was negatively correlated with the salivary testosterone levels among male
participants, that is, androgen status ($r = -0.20$, $p = 0.08$). The same trend was observed
between RASISB and testosterone ($r = -0.25$, $p = 0.03$).

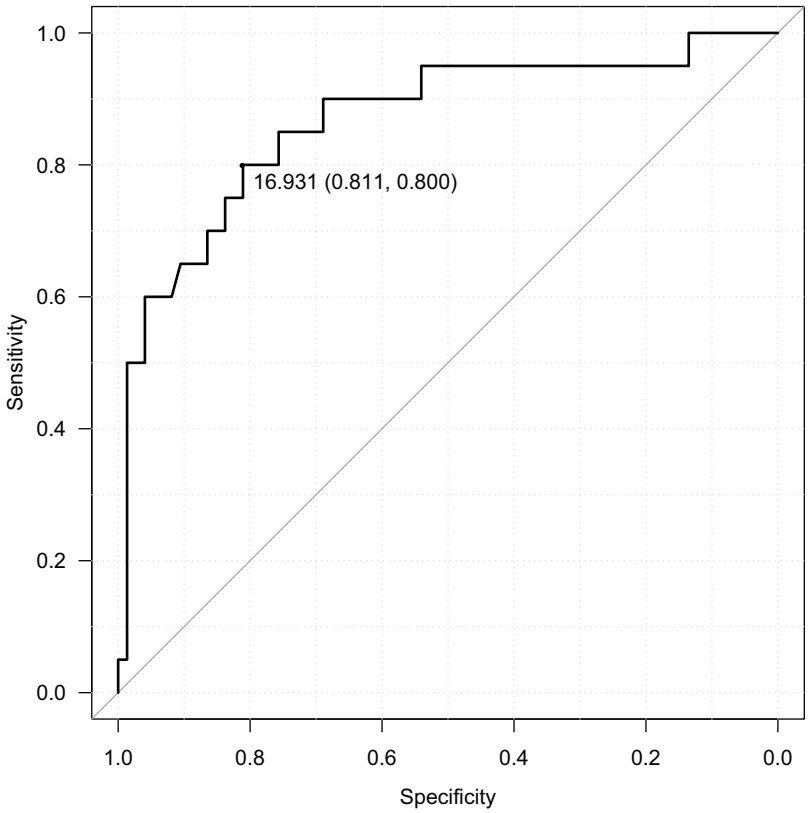

**Figure 5 ROC curve of adulthood RBIB discrimination proprieties of adolescent obesity (males).** The black dot represents an optimal RBIB cutoff value for distinguishing the history of obesity (16.9) with high sensitivity (0.80) and specificity (0.81).

For simultaneous evaluation of the relationships between body fat (both adolescent and adulthood) and sex hormones on one side and RBIB on the other side, we have used a multivariate regression with a reduction of dimensionality (OPLS model) and ordinary multiple regression (Table 2).

Using OPLS, the RBIB predictive component was strongly positively correlated with both adolescent BF% ($R^{\alpha}$ = 0.90, $p < 0.01$) and adulthood BF% ($R^{\alpha}$ = 0.81, $p < 0.01$). There was also a negative correlation between RBIB with salivary testosterone ($R^{\alpha}$ = −0.42, $p < 0.05$) but a positive one with estradiol to testosterone ratio ($R^{\alpha}$ = 0.50, $p < 0.01$). The model explained 54.8% variability of RBIB (Fig. 7).

## DISCUSSION

The present study showed a strong relationship between the amount of fat tissue in adolescence and relative pelvic breadth in early adulthood among both sexes. This trend was clearly illustrated in adults with a history of obesity, which has significantly broader pelvis in relation to body height (RBIB and RASISB) in comparison to individuals without a history of obesity. In this context, pelvic breadth can be used in adulthood as a highly sensitive and specific marker of a history of obesity during adolescence. On the male subset was observed weak negative relationship between relative pelvic breadth and

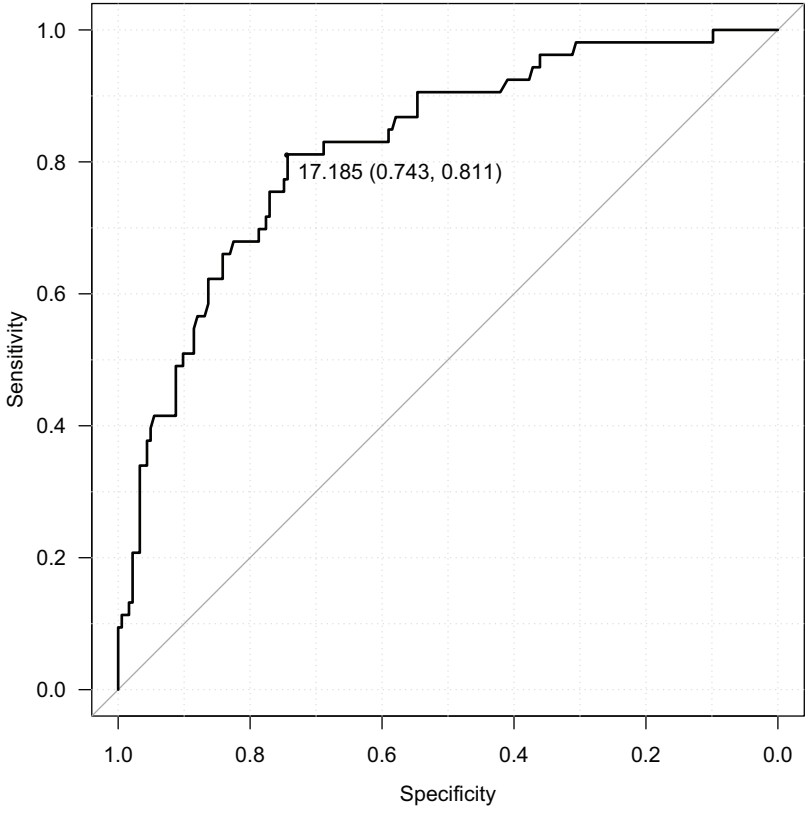

**Figure 6 ROC curve of adulthood RBIB discrimination proprieties of adolescent obesity (both sexes).** The black dot represents an optimal RBIB cutoff value for distinguishing the history of obesity (17.2) with high sensitivity (0.81) and specificity (0.74).

**Table 2 Relationships between RBIB and predictors as evaluated by OPLS model (for details see "Statistical analysis").**

| | Variable | OPLS | | | Multiple regression | |
|---|---|---|---|---|---|---|
| | | Component loading | $t$-statistics | $R^a$ | Regression coefficient | $t$-statistics |
| Relevant predictors (matrix $X$) | Adolescent BF% | 0.589 | 10.53 | 0.809** | 0.320 | 10.89** |
| | Adulthood BF% | 0.659 | 8.30 | 0.904** | 0.386 | 9.10** |
| | T | −0.306 | −2.05 | −0.422* | −0.144 | −2.32* |
| | E2/T | 0.379 | 4.37 | 0.501** | 0.140 | 3.59** |
| (matrix $Y$) | RBIB | 1.000 | 9.18 | 0.740** | | |
| Explained variability | | 54.8% (53.4% after cross-validation) | | | | |

Notes:
$R^{\alpha}$, Component loadings expressed as a correlation coefficients with predictive component.
* $p < 0.05$.
** $p < 0.01$.
$t$-statistic represents a ratio of component loading and statistical error.
All variables included in the initial OPLS model were relevant.

testosterone using basic linear models. Using a complex OPLS model analyzing the relationship between body fat (adolescent and adulthood), testosterone and estradiol to testosterone ratio on one side and RBIB on the other side we found a significant and relevant positive link between body fat, estradiol to testosterone ratio and RBIB;

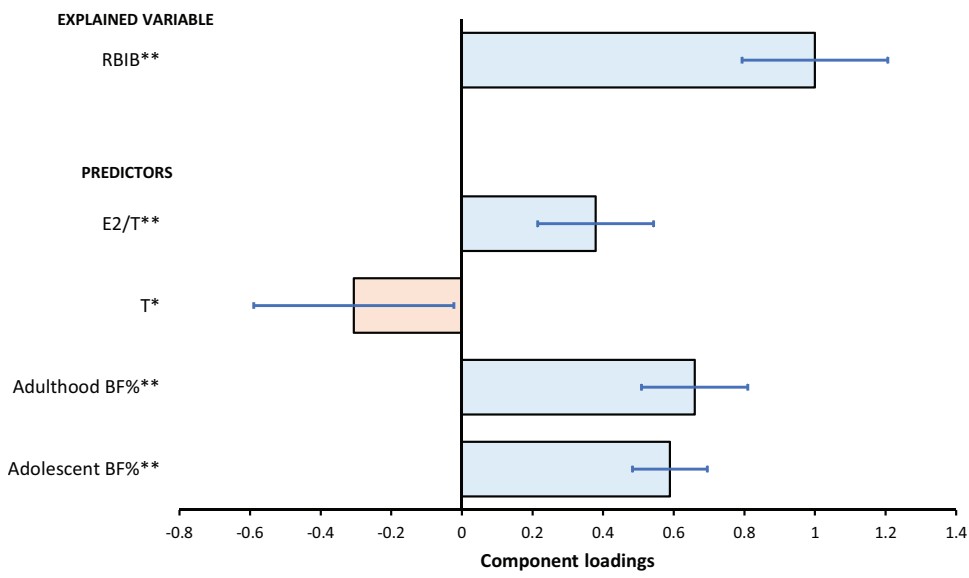

**Figure 7 The relationships between RBIB and predictors as evaluated by OPLS model.** The results of the OPLS model for predicting RBIB (Explained Variable) based on T, E2/T, adolescent and adulthood BF% (Predictors). The error bars represents 95% confidence interval of the parameters (component loadings expressed as regression coefficients); asterisks indicates the statistical significance: $^*p < 0.05$, $^{**}p < 0.01$.

testosterone was negatively related to RBIB and this model was able to describe over 54% of RBIB variability among males.

In the field of anthropology, RBIB is generally used to estimate the total mass of an individual from skeletal remains (*Ruff, Trinkaus & Holliday, 1997*). The estimation of body mass was validated on marine recruits, developing country population (Karkar Island), and also athletes (*Ruff, 2000*). The validation study performed on the recent general population indicates that this method can successfully predict the total body mass in normal-weight individuals but is significantly inconsistent among individuals with higher BMI (*Lorkiewicz-Muszyńska et al., 2013*). We assume that this inconsistency can be caused by differences in the body build based on the development of obesity. The body build and health status of an individual who became obese in adulthood are not the same as those who became obese during a period of extensive body development, such as adolescence.

Our results of the relationship between male pelvic breadth and sex hormones in the context of body fat indicate a possible link: a higher body fat leads to a higher rate of testosterone to estrogen conversion (adipose estrogenization). This higher conversion rate is expressed in lower testosterone levels and a higher E2/T ratio caused by the depletion of free circulating testosterone (*Fejes et al., 2006*). This is subsequently expressed in the relatively wider pelvic breadth, similarly to experiments with the administration of estrogens on animal models (*Hughes & Tanner, 1974*). There are only a few observational studies focused directly on the relationship between human pelvic breadth and steroid hormone levels, predominantly aimed at females. A later onset of puberty (based on the age of menarche) was related to lower values of RBIB in adulthood

(*Kirchengast & Hartmann, 1994*). A possible explanation of this phenomenon can be lower levels of estrogens related to a delayed onset of puberty. Also girls who exhibit lower values of estradiol had a narrower pelvis in comparison with the average weight females of the same age (*De Ridder et al., 1993*). In consistency with this, we observed that male adolescents with higher body fat and with that related elevated levels of estrogens tend to have a relatively wider pelvis and lower overall androgenic status in adulthood.

We assume that the influence of adipose estrogenization on the pelvic breadth in the context of obesity is an additive factor to the influence of greater biomechanical load and changed gait patterns. Obese individuals (including children) tend to have a wider step, lower cadence, and longer gait cycle compared to normal-weight individuals (*Hills & Parker, 1991*; *Ko, Stenholm & Ferrucci, 2010*). These changes are probably adaptations to a higher biomechanical load and together can lead to a change in morphology of femur with possible implications for pelvis (*Agostini & Ross, 2011*; *Harrington & Wescott, 2015*). In addition, estrogens have been discussed as a factor positively influencing mechanosensitivity of bone tissue (*Devlin, 2011*), obesity-related estrogenization can lead to even more significant response of skeletal structures to biomechanical stress.

Many studies observed not only a secular trend in increasing of the body height but also a secular trend of increasing rates of obesity together with pelvis widening across the globe, whether in the Poland (*Woronkowicz et al., 2016*), China (*Zhang & Wang, 2009*) or Saudi Arabia (*Al-Hazzaa, 2007*). These observations are interesting in the context of our findings. We assume that this increase is indicative of a global change in the human skeleton with a main suspected factor—obesity during childhood. Nevertheless, it is also necessary to reflect other possible factors such as changes in biomechanical stress and locomotion.

### Limitations

Due to the retrospective design of the present study, pelvic breadth was not obtained during enrollment. Consequently, it was impossible to link pelvic breadth from adolescence to pelvic breadth in adulthood. Different instruments were used to estimate fat; despite the same manufacturer and method being involved, the results may not be completely comparable.

For the ROC analysis, obesity was classified using BMI and BMI-SDS method. High BMI does not directly implicate high adiposity (BF%) (*Meeuwsen, Horgan & Elia, 2010*), even that this relationship was in the case of our study relatively high. Classification of obesity using BMI (and BMI-SDS) is not able to capture variations in body composition, on the other hand, there are no internationally accepted standards for classifying obesity based on BF% among adolescents.

## CONCLUSIONS

The level of adiposity (body fat) in adolescence is reflected in the skeletal build in adulthood. Adult individuals with a history of obesity and associated higher adiposity have a relatively wider pelvis to body height compared to individuals without a history of obesity and normal adiposity.

A possible explanation for this phenomenon may be adipose estrogenization and the associated habitual feminization. In this context, adult men with relatively broader pelvis and with that connected elevated adiposity tended to have lower testosterone levels and elevated estradiol to testosterone ratio.

Future research should focus on the morphology of other sexually dimorphic skeletal structures. It would be useful to use methods with a deeper explanatory value such as geometric morphometry than direct anthropometry and also to analyze a wider range of steroid hormones.

## LIST OF ABBREVIATIONS

| | |
|---|---|
| **ASISB** | Bi-spinal pelvic breadth (cm) |
| **AUC** | Area Under Curve |
| **BF%** | Body fat percentages measured using BIA |
| **BIA** | Bioelectrical Impedance Analysis |
| **BIB** | Bi-iliac pelvic breadth (cm) |
| **BMI** | Body Mass Index (kg/m$^2$) |
| **BMI-SDS** | Body Mass Index Standard Deviation Score |
| **E2** | Salivary estradiol (pg/mL) |
| **E2/T** | Salivary estradiol to testosterone ratio |
| **ELISA** | Enzyme-Linked Immunosorbent Assay |
| **OPLS** | Orthogonal Predictions to Latent Structure |
| **RASISB** | Relative bi-spinal pelvic breadth (bi-spinal breadth (cm) to body height (cm) ratio × 100) |
| **RBIB** | Relative bi-iliac pelvic breadth (bi-iliac breadth (cm) to body height (cm) ratio × 100) |
| **ROC** | Receiver Operating Characteristic |
| **T** | Salivary testosterone (pg/mL) |

## ACKNOWLEDGEMENTS

We would like to thank all participants for their willingness to participate in this study.

### Funding

This work was supported by the Charles University Grant Agency (GA UK No. 1250317) and the Czech Ministry of Health (AZV No. 17-31670A). The funders had no role in study design, data collection and analysis, decision to publish, or preparation of the manuscript.

### Grant Disclosures

The following grant information was disclosed by the authors:
Charles University Grant Agency: 1250317.
Czech Ministry of Health: 17-31670A.

## Competing Interests

The authors declare that they have no competing interests.

## Author Contributions

- Jan M. Novak conceived and designed the experiments, performed the experiments, analyzed the data, prepared figures and/or tables, authored or reviewed drafts of the paper, and approved the final draft.
- Jaroslav Bruzek conceived and designed the experiments, authored or reviewed drafts of the paper, and approved the final draft.
- Hana Zamrazilova conceived and designed the experiments, authored or reviewed drafts of the paper, and approved the final draft.
- Marketa Vankova conceived and designed the experiments, authored or reviewed drafts of the paper, study Coordinator, and approved the final draft.
- Martin Hill conceived and designed the experiments, analyzed the data, prepared figures and/or tables, authored or reviewed drafts of the paper, and approved the final draft.
- Petr Sedlak conceived and designed the experiments, authored or reviewed drafts of the paper, and approved the final draft.

## Human Ethics

The following information was supplied relating to ethical approvals (i.e., approving body and any reference numbers):

The Institutional Review Board of the Faculty of Science, Charles University in Prague approved this study (IRB Approval No. 201719).

## Data Availability

All analyzed data including error values and outliers are available in the Supplemental Files.

## Supplemental Information

Supplemental information for this article can be found online at http://dx.doi.org/10.7717/peerj.8951#supplemental-information.

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
