# Peer review of "The relationship between adolescent obesity and pelvis dimensions in adulthood: a retrospective longitudinal study"

_PeerJ, doi:10.7717/peerj.8951_

## Round 0.1 · original submission · Major Revisions

The three reviewers and I agree that the research is a valuable contribution to the field and is appropriate for publication in PeerJ. There are, however, some major revisions suggested that will strengthen the manuscript. First, all reviewers comment that the introduction needs to be expanded to provide more context. Please see each separate review for suggestions and missing information. Second, two reviewers note that the discussion does not address alternate interpretations of the data sufficiently and I encourage you to read those comments and consider expanding the discussion as requested. Third, two of the reviewers note that the conclusions are over-stated given the limitations of the study and the other possible explanations for the findings. Please address this concern in your revision. Finally, clarify methods and terms as well as provide additional information for tables and graphs as noted in specific reviews.

I realize that the number of additional references suggested, particularly for the introduction, may seem challenging. Focus on providing more detail on suggestions that are found in 2+ reviews and less detail, if warranted, on suggestions found in only one review. In addition, there is a central concern over BMI and the factors that affect it, including muscle versus fat, the use of adiposity, and age at onset of high BMI. Clarifying these terms across the paper will provide strength to the analysis. There are also reviewer specific minor comments that should be addressed in the revision.

Reviewer 1 ·

Basic reporting

The article is mostly written in clear English throughout, and the literature review is thorough. The structure conforms to PeerJ standards. Figures are relevant, but some could be improved for clarity. The raw data Is supplied.

See below for specific comments:

Line 22 - Is "pathogenic" a necessary descriptor here?
Line 25 - add descriptor for "final height" - reduced? - spell out DM2, and "afterward lost their weight" is an awkward phrasing in English - subsequently lost weight?
Line 27 - delete "however, does" replace with "is"
Line 28 - delete "fully apply in", change "case of" to "case in" or "case for"; add "skeletal" before "differences"
Line 38 - add "(BIA)" after analysis
Line 49 - qualify what you mean by "satisfactory sensitivity..."
Line 53 - add "and gait pattern" after "activity"
Line 57 - Do you really want to start the paper with a definition from WHO? I suggest laying out the problem or question briefly with a strong impact in the beginning of your introduction
Lines 64-65 - I would delete the sentence "In comparison to individuals...getting married" as this is not extremely relevant and you don't mention it further.
Line 66 - add "of comorbidities" after "overview"
Line 70 - awkward wording for "individual's overweight..." maybe just delete "overweight", because it is the adjective and "obesity" is the noun
Line 73 – delete “another” replace with “a”
Line 75 – this is not exclusively a holistic perspective from toxicology – maybe generalize this to “perspective of a pathologist” – and it is problematic to think of fat and fat storage as purely pathological.
Line 84 - see comment for line 25
Line 108 – specify development in juveniles
Line 110 – should be “epiphyseal”
Line 118 and 119 – define ICIC a little better and refer to it also as the osteometric measurement of “bi-iliac breadth”
Line 124 – delete “on the” and replace with “in”
Line 125 – add “s” to be “animal models”
Line 145 – change “Caucasian Population” to “European ancestry”
Line 150 – specify number of females and males, instead of percentage of females – this is misleading
Line 152 – type – Tanita – second time – and include hyphen between SC-240
Line 153 and 154 – Should this be Salimetrics, LLC. This is what their website says.
Line 161 – Why are you not also using BIA for level of obesity/adiposity here?
Line 174 – Specify data is “nonparametric” if using Mann-Whitney and Spearman’s rank
Line 185 – Again, why not using BIA?
Line 198 – “The relationship was stronger for recent adiposity”? this seems like the reverse of what you found. Please explain this variable.
Lines 203-205 – Is there bias when conducting anthropometrics with higher versus lower adiposity? How can you ensure that soft tissues do not have an effect on the measure in terms of tissue depth?
Line 209 and 211 – spell out AUC
Line 216 – explain this finding more
Line 221-222 – “illustratively exhibited” is a little redundant – maybe “clearly illustrated”
Line 227 – “retrospective anthropology”? just say “anthropology”
Line 236 – add “approach” before “falls”
Line 238 – Delete “of an image”
Line 248 – Normal population – are you comparing to both males and females? If not then perhaps just “average weight females”
Lines 248 – 251– the sentence starting “In consistency with this, we assume…” Assume? Didn’t you just demonstrate/test this hypothesis?
Lines 253 – 255 – is this also associated with the same rate of increase in obesity rates?
Line 256 – “alarming” – this is dramatic – Other research may find extremely deleterious effects of obesity, but higher estradiol and wider pelvis are hardly life threatening.

Table 1 – the first 2 rows are not listing the age during enrollment and follow up. Additional columns would resolve this issue, as opposed to having these as extra rows.

Somewhere in the discussion, it should be addressed that gait changes associated with obesity can lead to changes in bone shape. Obese subjects walk with a more side-to-side saunter which appears to cause forces to change the shape of the bones of the femur and presumably the pelvis. Here are some references that deal with these changes in bone shape.

Agostini GM, Ross AH (2011) The effect of weight on the femur: a cross-sectional analysis. J Forensic Sci 56:339–343

Elliott M, Kurki H., Weston, DA, Collard, M., (2015) Estimating body mass from postcranial variables: an evaluation of current equations using a large known-mass sample of modern humans, Archaeol Anthropol Sci, DOI 10.1007/s12520-015-0251-6

Hills, AP, EM Hennig, NM Byrne and JR Steele. 2002. The biomechanics of adiposity-structural and functional limitations of obesity and implications for movement. Obes Rev 3:35-43.

Lacoste Jeanson A, Santos F, Villa C, Dupej J, Lynnerup N, Brůžek J, (2017) Body mass estimation from the skeleton: An evaluation of 11 methods. Forensic Sci Int. Dec;281:183.e1-183.e8. doi: 10.1016/j.forsciint.2017.10.026.

Moore MK, Schaefer E (2011) A comprehensive regression tree to estimate body weight from the skeleton. J Forensic Sci 56: 1115–1122

Moore MK. 2008. Body mass estimation from the human skeleton. PhD Dissertation. University of Tennessee, Knoxville.

Moro M, Van der Meulin MCH, Kiratli BJ, Marcus R, Bachrach LK and Carter DR. 1996. Body mass is the primary determinant of midfemoral bone acquisition during adolescent growth. Bone 19(5):519-526.

Experimental design

The research falls within the aims and scope of journal, the research question is well defined, the research is rigorous and methods apt.

Validity of the findings

The conclusions appear sound. There could be other confounding variables in terms of gait changes which are mentioned at the end of the "basic reporting" section.

Reviewer 2 ·

Basic reporting

The paper needs to be edited. As presented there are areas that are unclear and ambiguous.
The literature references is minimally sufficient.
The hypotheses are stated but the data is over interpreted.
The research appears to be original and within the aims and scope of the journal

Experimental design

The methods do not have sufficient detail to replicate

Validity of the findings

See general comments below. I believe the data may be over interpreted and there is a lack of discussion of other possible causal factors

Additional comments

This is an interesting and important study that examines the relationship between adult pelvic breadth relative to stature and adiposity. It is a retrospective study, so the research design does not allow the researchers to determine the causation of the correlations. While a limiting factor, especially the lack of hormonal data for females, it does not take away from the study. There are, however, some problems with the presentation of the research that needs to be addressed, especially since there are limitations associated with the retrospective study. For the methods there is very little information about how they controlled for body fat in the measurements, how stature was measured, how sensitivity and specificity were calculated, the BMI of the individuals during the original and follow up study. This information is necessary for the reader to understand the interpretation of the results. Regarding the interpretation, the main problem with this study is that there are a lot of assumptions and a lack of discussion of other possible causal factors. For example, there needs to be some discussion of normal pelvic width increases with age in adulthood and the difference in growth between males and females. For example, when does pelvic breadth complete growth in males and females and is the interpretation affected by the age difference between the males and female in their sample? Furthermore, stature is reduced in obese adolescents, at least among females, indicating that obese females should have a higher pelvic breadth/stature ratio even if the pelvis is not widening. In the graphs there is no indication of which individuals were of normal BMI or obese. For the males in your sample, there is little correlation with testosterone and relative pelvic breadth if males with extreme testosterone are removed. Again, were the individuals with high testosterone of normal BMI or where they obese? In addition , I am not sure you can make the broad argument that this study shows the limits of forensic anthropological methods without giving specific details. I assume the authors are referring to the use of pelvic breadth and stature to estimate body mass. However, the papers they cite are not forensic anthropological but rather paleontological where the goal is to reconstruct lean body mass when there were unlikely very few obese individuals.
Line 103 – what does this topic sentence have to do with the rest of the paragraph? There should be a separate paragraph that discusses the effects of low testosterone. Much of the paper needs to be edited for clarity and flow.
Table 1 – information on BMI of sample. It might also be interesting to know how many of the initial participants that were obese remained obese as adults
Figures 4-6 – no red dot

Reviewer 3 ·

Basic reporting

Generally, I found this paper was well-written and the author's main points were clear and unambiguous.

The table requires substantial revision (details in notes to authors)

Experimental design

The question is well-defined and relevant and makes a contribution linking hormones and skeletal morphology, although this manuscript would benefit from further background regarding endocrinology, inflammation, and obesity.

Validity of the findings

Conclusions are generally valid, but some of the theoretical linkages may be tenuous. Further details are below in comments to the authors.

Additional comments

In this paper, the authors examine the relationship between obesity during adolescence and pelvic breadth, based on the relationship between adipose tissue and estrogenic activity. They found that adiposity (measured with bioimpedance) was related to pelvic breadth in adulthood, and pelvic breadth in adulthood was a retrospective marker of obesity during adolescence. They did not find a significant association between salivary reproductive hormones and pelvic breadth.

Overall, I think it would be an interesting addition to the literature about how conditions during growth and development shape adult health and skeletal phenotype, but it requires a few clarifications and a bit more analyses.

In general:
The authors switch between using the terms “obesity” and “adiposity”, and at times this is confusing, as adiposity seems to be the theoretical driver of hormonal changes that affect the pelvis. In comparison, the author’s definition of obesity is unclear (what is the threshold for obesity?) and I’m not sure if in the current form it adds to the main narrative and argument presented here.
I would recommend either further developing the model presented in the introduction (more detail in background and discussion) or removing it.

Specific recommendations:
1. Abstract
a. The abstract describes collecting salivary samples for measurement of testosterone and estrogen, but estrogen was not reported in results?
b. “Anamnesis” is an unusual term to use here. I assume the authors use it to refer to medical history, but they then don’t report any kind of medical history?
c. Estrogen is mentioned in the typeset Abstract Background, but not in the draft (lines 39-40), and it is not included in the rest of the manuscript.

2. Introduction
a. I would recommend the authors also include some references on how social stigma (such as incurred by obese people in a fat-shaming world) can affect inflammatory and stress pathways. Some of the physical and psychological effects of obesity are likely linked to these social experiences of bullying and exclusion.
b. The model proposed in lines 73-78 about analogizing obesity as a toxic substance is not really developed further or revisited in the rest of the manuscript, so I’m not sure why it is included here.
c. In line 100-101, the authors mention fat tissue being correlated with the level of estrogens, but then both references focus on estrogens in men. Perhaps include some references about healthy adult women, such as Ziomkiewicz et al (2008), or focus this manuscript on males?
d. Given that this isn’t a specialty journal, a diagram of the ICIC measurement might be useful for some readers.

3. Materials and Methods
a. Line 145: The Czech sample is likely better described as “white” or “Central European” instead of “Caucasian”, although it is not clear if the sample was screened for racial or ethnic identity?
b. Line 149: I’m unsure what probands you are referring to?
c. Lines 156-158: It would be helpful to provide additional detail regarding collection of salivary testosterone. How long after eating/drinking/brushing teeth? Was gum used to stimulate salivary production? Was the saliva collected before, during, or after the rest of the examination?
d. Lines 161-163: Please provide the thresholds for BMI standard deviation scores so we know the cutoffs you used for classifying “obesity” in this sample.
e. Table 1 needs to be reformatted so the row and column labels are accurate.
4. Results
a. Descriptive statistics correlating adolescent body fat and adolescent BMI would be useful, as from the reported information it’s clear you could test whether adiposity and BMI are related vs whether increased BMI is associated with increased musculature.
b. As part of the descriptive statistics, I would be interested to know whether people who were obese or had high adiposity as teenagers maintained that status into adulthood
c. Lines 202-211: Without a clear definition of the BMI threshold for obesity, or any understanding about whether high BMI was associated with high adiposity, I’m unclear about whether the ICICH ROC analyses are useful. I wonder what would happen if the analyses were performed in reverse…correlating adolescent BMI and ICICH, then constructing a ROC curve for ICICH and adiposity?
d. Lines 204-206: Please provide sample size for the number of individuals with and without a history of obesity. This could be provided as part of a revised table 1, or in text.
e. Lines 213-216: your subtitle says “ICICH as a marker of estrogenization”, but estrogen isn’t measured? Based on Figure 7, I’m not sure if a linear regression is the best method for examining the relationship between testosterone and ICICH.
5. Discussion
a. Further development of the author’s arguments regarding estrogen and testosterone in adolescents and men is needed, bolstered by references from endocrinology. I’m unclear that increased estrogens are necessarily associated with reduced testosterone, and it’s been shown that testosterone is converted to estrogen in bone tissue. This means there are several theoretical and biological linkages that could be strengthened in the argument between BMI --> adipose tissue --> estrogen --> testosterone/bone morphology.
b. Line 256: I’m unsure why the authors find the observation of increased pelvis breadth “alarming” based on this discussion.
c. Finally, please spend a bit more time discussing the potential differences between high BMI (which can be caused by body fat or muscle mass) and high adiposity.


Ziomkiewicz, A., P. T. Ellison, S. F. Lipson, I. Thune, and G. Jasienska. “Body Fat, Energy Balance and Estradiol Levels: A Study Based on Hormonal Profiles from Complete Menstrual Cycles.” Human Reproduction 23, no. 11 (2008): 2555–63.

---

## Round 0.2 · Minor Revisions

The revised manuscript was deemed to be significantly improved and, pending a few minor revisions, ready for publication. An ongoing editorial concern is the acroynums. There are many acronyms to keep track of in this manuscript. Some are common (BMI) but others are not, nor are they referenced in a logical manner. For example, I am not clear on how bi-iliac breadth is abbreviated as ICIC (or how that relates to the undefined ICICH). I-I or BIB would make more sense. Please consider making this less commonly used acronyms more intuitive for the reader. Please address the few remaining reviewer comments in your revision.

Reviewer 2 ·

Basic reporting

no comment

Experimental design

no comment

Validity of the findings

no comment

Additional comments

Greatly improved and interesting study.
Line 60 - delete "yet"
Lines 71-77 - is this really necessary? it is not discussed again nor does it seem to influence your results.
Line 127 - Sentence "However, in the case of the juvenile pelvis, it does to fully apply" is a little confusing. Try just adding it to the previous sentence so it reads something like "...adult pelvis is sexually dimorphic while the juvenile pelvis is not."
Line 129 - morphological differences BETWEEN SEXES are distinct.
Line 151 - You state the next stage is to validate if relative pelvic breadth can be used as a classifier of the history of obesity. It is still unclear as to why you want to predict if someone was obese as a juvenile. The research question is does obesity during adolescence influence the pelvic breadth to stature ratio? But why predict obesity as a juvenile? Couldn't you just ask the patient? This could be corrected with a single sentence.
Line 189 - replace "bi-iliac" with "bi-spinal." You state in 186 that bi-iliac breadth can be influenced by soft tissue and then on 189 say that the advantage of bi-iliac breadth is that there is not interaction with soft tissues.
Line 245 - You never define ICICH in the text. You could go back to line 190 and add them so it looks something like Both pelvic dimensions were analyzed in relation to body height and indicated as ICICH and ASISH.

Reviewer 3 ·

Basic reporting

No additional comments

Experimental design

No additional comments

Validity of the findings

No additional comments

Additional comments

The authors have mostly addressed my concerns and I feel the manuscript is generally acceptable.

In Figure 7 the asterisks (*) are not explained in the figure legend or text. Additionally it might be more effective as a boxplot instead of a bar chart, since I believe it represents an estimate and range, rather than count data.

---

## Round 0.3 · accepted · Accept

Thanks very much for the resubmission and addressing the final issues with the manuscript. I think this will make a nice contribution across multiple fields.